# Stored Energy Increases Body Weight and Skeletal Muscle Mass in Older, Underweight Patients after Stroke

**DOI:** 10.3390/nu13093274

**Published:** 2021-09-19

**Authors:** Yoshihiro Yoshimura, Hidetaka Wakabayashi, Ryo Momosaki, Fumihiko Nagano, Takahiro Bise, Sayuri Shimazu, Ai Shiraishi

**Affiliations:** 1Center for Sarcopenia and Malnutrition Research, Kumamoto Rehabilitation Hospital, Kumamoto 869-1106, Japan; f-nagano@kumareha.jp (F.N.); asian.dub.foundation00@gmail.com (T.B.); shimazu@kumareha.jp (S.S.); ai.shiraishi0913@gmail.com (A.S.); 2Department of Rehabilitation Medicine, Tokyo Women’s Medical University Hospital, Tokyo 162-0054, Japan; noventurenoglory@gmail.com; 3Department of Rehabilitation Medicine, Mie University Graduate School of Medicine, Tsu 514-8507, Japan; momosakiryo@gmail.com

**Keywords:** stored energy, body weight gain, skeletal muscle mass gain, malnutrition, aggressive rehabilitation nutrition

## Abstract

We conducted a retrospective observational study in 170 older, underweight patients after stroke to elucidate whether stored energy was associated with gains in body weight (BW) and skeletal muscle mass (SMM). Energy intake was recorded on admission. The energy requirement was estimated as actual BW (kg) × 30 (kcal/day), and the stored energy was defined as the energy intake minus the energy requirement. Body composition was measured by bioelectrical impedance analysis. The study participants gained an average of 1.0 ± 2.6 kg of BW over a mean hospital stay of 100 ± 42 days with a mean stored energy of 96.2 ± 91.4 kcal per day. They also gained an average of 0.2 ± 1.6 kg of SMM and 0.5 ± 2.3 kg of fat mass (FM). This means about 9600 kcal were needed to gain 1 kg of BW. In addition, a 1 kg increase in body weight resulted in a 23.7% increase in SMM and a 45.8% increase in FM. Multivariate regression analyses showed that the stored energy was significantly associated with gains in BW and SMM. Aggressive nutrition therapy is important for improving nutritional status and function in patients with malnutrition and sarcopenia.

## 1. Introduction

Malnutrition is commonly observed in the geriatric population and is associated with adverse outcomes in geriatric rehabilitation patients. Stroke or hip fracture patients are often transferred from acute-care hospital wards to convalescent rehabilitation wards, and these patient groups show a high prevalence of malnutrition and sarcopenia at 49–67% [1,2] and 40–53% [3,4], respectively. Malnutrition, weight loss, body mass index (BMI) lower than 20 kg/m^2^, sarcopenia, and reduced nutritional intake are established independent factors that negatively influence functional recovery in older inpatients [5,6,7]. The goal of geriatric rehabilitation is to promote functional recovery and thereby allow hospitalized patients to return to their homes. Therefore, it is important to improve the nutritional status and sarcopenia in older rehabilitation patients to maximize favorable outcomes.

Aggressive nutrition therapy improves the outcomes of older hospitalized patients [8,9,10]. Weight gain and muscle mass gain during hospitalization have a positive effect on activities of daily living (ADL) in these patients [11,12]. In older patients with low BMI or sarcopenia, daily energy expenditure and energy stores need to be considered in terms of energy needs. Energy expenditure in healthy individuals consists of resting metabolic rate, diet-induced thermogenesis, and energy expenditure related to activity and illness. On the other hand, for patients who are malnourished and for those with sarcopenia, energy requirements need to be estimated by adding stored energy for the restoration of lean body mass [8]. To gain 1 kg of body weight (BW), individuals aged 10–40 years need 7500 kcal [13,14].

However, there is a lack of evidence on the relationship between stored energy and weight gain and skeletal muscle mass (SMM) gain in geriatric rehabilitation patients. Hence, we conducted a retrospective cohort study to determine whether the stored energy intake was associated with gains in BW and SMM, as well as how much energy was needed to gain 1 kg of BW in older patients with low BW undergoing convalescent rehabilitation after stroke.

## 2. Materials and Methods

### 2.1. Study Design and Participants

We conducted a retrospective cohort study at a post-acute care hospital, which included convalescent rehabilitation wards with a total of 135 beds [15]. The study was conducted between January 2016 and December 2020 and included all stroke patients newly admitted to the wards who were over 70 years old and had a BMI of less than 20.0 kg/m^2^ [16]. Patients were excluded from this study if they refused to participate, had incomplete data, or had edema of the extremities, pleural or ascites fluid, or altered consciousness. The observation period lasted from the date of admission to the date of discharge.

### 2.2. Data Collection

The baseline patient demographic characteristics were recorded upon admission, including age, sex, body mass index, stroke type, stroke history, days from stroke onset to admission [17]; nutritional status, assessed using the Mini Nutritional Assessment-Short Form (MNA-SF) [18]; swallowing status, assessed using the Food Intake Level Scale (FILS) and presence of dysphagia requiring supplemental feeding defined by the FILS scores <7 [19]; comorbidities, assessed using the Charlson Comorbidity Index (CCI) [20]; and premorbid ADL, assessed using the modified Rankin Scale (mRS) [21]. Information was collected on the presence of paralysis and localization and stage of paralysis according to the Brunnstrom recovery stages (BRS) [22]. Blood sampling data on admission, including albumin, hemoglobin, and C-reactive protein levels were recorded. The number of drugs prescribed at the time of admission was recorded. Data on the total rehabilitation therapy received during hospitalization (units per day, 1 unit = 20 min of therapy) were extracted from medical records.

Within 72 h of admission, bioelectrical impedance analysis (BIA) data of the SMM and fat mass (FM), hand grip strength (HG), and functional independence measure (FIM) scores for physical and cognitive function (FIM-motor and FIM-cognitive) [23] were obtained. FIM gain was calculated by subtracting the FIM score at discharge from the FIM score at admission. The BIA measurements were carried out using the latest version of a validated instrument (InBody S10: InBody, Tokyo, Japan) and following a standard protocol explained elsewhere [24]. SMM and FM each were divided by the square of height (m) and indexed as the skeletal muscle mass index (SMI) and fat mass index (FMI). HG was measured using the Smedley hand dynamometer (TTM, Tokyo, Japan) in the non-dominant hand (or in case of hemiparesis, in the non-paralyzed hand), with the patient in the standing or seated position (depending on their ability) and with their arms relaxed at their side; measurements were taken three times, and the highest value was recorded. Sarcopenia was diagnosed when both the SMI and HG values were low, in accordance with the Asian Working Group for Sarcopenia criteria 2019 [25].

### 2.3. Stored Energy and Nutritional Intakes

Energy and protein intakes were calculated by a nurse or dietitian who visually assessed the ratio of intake to the amount provided to the patient. The intake from three servings each of breakfast, lunch, and dinner (a total of nine servings) was recorded [26], and the average of each value divided by three was used as the daily intake. In the case of enteral (EN) and parenteral nutrition (PN), the energy and protein doses for the first 72 h of hospitalization were recorded, and the respective values divided by 3 were used as daily intake. If oral intake was combined with EN or PN, the respective energy and protein intake (administered) were added up. In addition, nutrient intake was calculated by dividing each intake by the actual body weight on admission. Nutritional intake was recorded at admission and at discharge.

The energy requirement for older patients undergoing stroke rehabilitation was estimated as actual BW (kg) × 30 (kcal/day) [8,27], and the stored energy was defined as the energy intake minus the energy requirement. According to the stored energy on admission, patients were divided into two groups: those whose stored energy was greater than 0 kcal/day and those whose stored energy was less than 0 kcal/day.

### 2.4. Outcomes

The primary outcome was BW gain, which was defined as the change in BW during hospitalization (BW at discharge—BW at admission). The secondary outcome was SMM gain, which was defined as the change in SMM during hospitalization (SMM at discharge—SMM at admission).

### 2.5. Convalescent Rehabilitation

The convalescent rehabilitation program (up to 3 h per day) was performed according to the guidelines of the National Health Insurance System. The program was tailored to suit the functional abilities and disabilities of the patient, such as physical therapy including paralyzed limb facilitation (for leg paralysis), range of motion exercises, basic movement training (mainly for the legs), walking training, resistance training (such as chair-stand exercises [28]), and ADL training [29].

For nutritional management, nutritional screening and nutritional assessment were conducted for all patients, and under the guidance of the dietitians and nutrition support team, active nutritional support was provided, including high-energy and/or high-protein meals. In addition, nutrition management was tailored to each patient’s condition and nutritional needs by adjusting energy and protein according to changes in rehabilitation time and load [30].

Dysphagia rehabilitation was customized to the patients’ swallowing abilities and function, and included oral management and exercise, indirect (without food) and direct (with food) exercises, and diet modification through multi-occupational collaboration with speech and swallowing therapists, dental hygienists, and ward staff [31].

Oral management included oral screening, assessment, education, counseling, treatment (oral care), oral and dysphagia rehabilitation, medical treatment by a dentist, and practicing in cooperation with a multidisciplinary team [31]. Ward dental hygienists conducted oral and dysphagia rehabilitation, including indirect and direct (oral intake) exercises at the patient’s bedside [32].

Medication management was carried out by multidisciplinary teams, including pharmacists. Pharmacotherapy is one of the factors that affect the nutritional state of older people. Polypharmacy and inappropriate medications were corrected, and medications that could affect nutritional status were managed throughout the hospital stay [33].

### 2.6. Sample Size Calculation

The sample size for statistical power was calculated by using data from our previous study [34], and the results showed that the BW of patients at hospital admission was normally distributed with a standard deviation (SD) of 8.0. If the true difference between the mean values of BW at discharge of patients with more and less stored energy is 4.0, a sample size of at least 64 participants would be required in each group to reject the null hypothesis with a power of 0.8 and an alpha error of 0.05, and this would support the validity of our results.

### 2.7. Statistical Analysis

The values were reported as the mean (SD) for parametric data or as the median (interquartile range; IQR) and numbers (%) for non-parametric and categorical data, respectively. In the univariate analyses, the patients were stratified according to the intake of stored energy (with or without stored energy). Comparisons between groups were made using the t-test, Mann-Whitney *U* test, and chi-square test, as suitable.

Multiple linear regression analysis was carried out to determine whether the stored energy was independently associated with the study outcomes, including BW gain and SMM gain during hospitalization. The covariates of age, sex, length of hospital stay, BRS of the lower limb, FIM-motor and FIM-cognitive scores at admission, total rehabilitation therapy (units/day), energy intake at baseline, FMI and SMI on admission, and stroke type, all of which are considered to be confounding factors affecting BW gain, were included in the model. Multicollinearity was assessed using the variance inflation factor (VIF): a VIF value of 1–10 indicated the absence of multicollinearity. Statistical significance was set at *p*-values of <0.05. All analyses were conducted using IBM SPSS version 21 (Armonk, NY, USA).

### 2.8. Ethics

This study was approved by the Institutional Review Board (approval no.: 169-210617) of the hospital where the study was conducted. Written informed consent could not be obtained because of the constraints imposed by the retrospective study design, although the participants could withdraw from this study at any time by using an opt-out procedure. This study was conducted in accordance with the Declaration of Helsinki and ethical guidelines for medical and health research involving human subjects.

## 3. Results

During the study period, a total of 843 stroke patients were newly admitted to the hospital. Patients with missing data (*n* = 22) and altered consciousness (*n* = 8) were excluded. Data from 813 patients were considered for inclusion in the study. Among them, patients aged 70 years or younger (*n* = 418) and those with BMI ≥ 20 (*n* = 583) were excluded. Finally, 170 patients who were over 70 years of age and had a BMI of <20.0 were included in the analysis (Figure 1).

The patients’ baseline characteristics are presented in Table 1. The mean age was 83.1 ± 6.2 years; 52% of the patients were male. The median BW and SMM were 41.6 kg (IQR: 38.2–44.1) and 10.9 kg (IQR: 8.8–13.2), respectively. The median MNA-SF was 4 (IQR: 3–6), suggesting that a large population of the patients was malnourished. The median FIM-motor and FIM-cognitive scores were 32 (IQR: 14–58) and 17 (IQR: 10–25), respectively, suggesting that a large proportion of patients was physically dependent at the baseline. Sarcopenia was observed in 84.7% of the patients (*n* = 144), with a median HG of 12.5 kg (IQR: 5.6–17.8) and SMI of 4.9 kg/m^2^ (IQR: 4.0–5.7).

Overall, the study patients gained an average of 1.0 ± 2.6 kg of BW over a mean hospital stay of 100.2 ± 42.3 days, with a mean stored energy intake of 96.2 ± 91.4 kcal per day. They also gained an average of 0.2 ± 1.6 kg of SMM and 0.5 ± 2.3 kg of FM, respectively. This means that it took about 9600 kcal (9614.2) for the patient to gain 1 kg of BW (mean stored energy per day [96.2] × mean number of days in hospital [100.2]/mean BW increase [1.0]). In addition, a 1 kg increase in body weight resulted in a 23.7% increase in SMM (mean SMM increase ÷ mean BW increase × 100) and a 45.8% increase in FM (mean FM increase ÷ mean BW increase × 100). Table 2 shows a two-group comparison of the nutritional profiles of patients with and without stored energy intake by stroke type. Except for patients with subarachnoid hemorrhage (SAH), patients with stored energy intake of more than 0 kcal/day consumed more energy and protein on admission and at discharge than those with stored energy intake of less than 0 kcal/day. All patients with SAH were supplemented with stored energy. In the univariate analysis, there was no difference in BW gain or SMM gain between the two groups.

In the multivariate linear regression analysis, the stored energy intake was significantly and positively associated with BW gain during hospitalization (*β* = 0.256, *p* = 0.005) after adjusting for potential confounders, including age, sex, baseline energy intake, FMI and SMI (Table 3). Moreover, the stored energy intake was significantly and positively associated with SMM gain during hospitalization (*β* = 0.263, *p* = 0.011) in the same analysis model (Table 4).

## 4. Discussion

In this study, we determined whether stored energy intake was associated with gains in BW and SMM during hospitalization in older, underweight, post-stroke rehabilitation patients, and highlight two important findings: (1) Stored energy intake was associated with gain in BW and SMM, and (2) it took about 9600 kcal to gain 1 kg of body weight in these patients.

Stored energy intake was associated with gain in weight and muscle mass during hospitalization in geriatric rehabilitation after stroke. The results of the univariate analysis showed differences by stroke type, but the small sample size of 12 patients with SAH might have affected the results. On the other hand, since we adjusted for stroke type in the multivariate analysis, we consider that the effect of stroke type on the results was controlled. Considering the fact that weight gain and increased muscle mass have a positive impact on the ADLs of these patients [11,12], our findings suggest the need for enhanced nutritional management for older rehabilitation patients with low body weight. However, strong evidence exists that a large proportion of stroke patients do not even meet their estimated energy requirement both in hospital and after discharge [35]. An increase in energy demand related to rehabilitation, chronic diseases, and improvement of sarcopenia and malnutrition would likely result in negative energy balance in geriatric rehabilitation. Therefore, it is important to ensure the provision of individualized stored energy for weight gain in addition to energy requirements, while energy calculations should take into account post-stroke paralysis, muscle spasticity, physical disabilities, nutritional status, and age and sex in geriatric rehabilitation after stroke. Furthermore, attention also needs to be paid to medications that are often prescribed to post-stroke patients. For example, statins are myotoxic and have been suggested to be associated with the progression of sarcopenia [36]. On the other hand, there is evidence that statin use reduces the incidence of sarcopenia, so if a patient is a good candidate for statin use, it is necessary to take care of myopathy and sarcopenia while using statins [37].

To increase 1 kg of BW, about 9600 kcal were required for older rehabilitation patients who were underweight. Positive energy balance is necessary for growth, wound healing, and muscle gain, but increased intake of energy or prolonged intake can lead to overweight and obesity [14]. Therefore, specific numerical evidence is needed for nutritional planning so as to achieve a positive energy balance that can promote weight gain in underweight patients. In the literature, to gain 1 kg of lean body mass, individuals aged 10–40 years needed 7500 kcal [13,14], while older individuals needed 8800–22,600 kcal [38]. The diversity of energy requirements for weight gain probably depends on the subject’s background, including age; sex; nutritional status; changes in body composition; the type, intensity, and duration of rehabilitation and exercise; presence of systemic inflammation; and comorbidities. In this study, we estimated the stored energy for weight gain in lower-weight older patients undergoing stroke rehabilitation. We expect this estimate to help improve the nutritional status of older patients in daily clinical practice. However, further studies are needed to validate our result.

Ensuring that underweight older patients obtain the required level of stored energy is a practical nutritional management. This may also be true for underweight older patients in general, not just stroke patients. Aggressive rehabilitation nutrition improves nutritional and functional outcomes in patients with malnutrition and sarcopenia [8]. Malnutrition and sarcopenia negatively affect functional recovery and ADL, while nutrition improvement, including weight gain and muscle mass gain, is positively associated with functional recovery [11,12]. Aggressive nutrition therapy described here is characterized by the setting of energy requirements based on the amount energy expenditure per day plus stored energy. This is not the same as enhanced nutrition therapy and instead focuses on improving malnutrition and sarcopenia. Moreover, rehabilitation nutrition care process suggests that aggressive nutrition therapy should be performed in combination with aggressive exercise therapy [39]. Medication is another factor that can affect the nutritional status in older patients undergoing rehabilitation. Since some of the most important drugs that affect nutritional status are those that cause anorexia, rehabilitation pharmacotherapy should be practiced [33]. Therefore, it is necessary to reconcile the rehabilitation goal setting, the content, quantity, and quality of physical activity and exercise therapy, and the patient’s general condition to set nutrition goals through multidisciplinary collaboration.

This study had some limitations. First, the daily stored energy intake was based on estimates at the time of admission and not on weight change over the course of the hospital stay. This could be a major limitation of this study, as energy adjustments are made as appropriate for weight changes over time in daily clinical practice. Moreover, determination of energy requirements by indirect calorimetry over time would provide more accurate estimates than those used in this study. Further, this study involved a single rehabilitation hospital in Japan, possibly limiting the generalizability of the findings. In addition, there was a selection bias in that the subjects were older hospitalized patients with low body weight. Future multicenter studies are needed to determine whether similar results can be obtained in diverse populations, especially by stroke type. Lastly, due to its retrospective study design, we were unable to obtain detailed information on nutritional therapy during hospitalization, possibly affecting the results. High-quality prospective studies adjusted for confounders are needed in the future.

## 5. Conclusions

Stored energy intake was associated with weight gain and muscle mass gain in underweight older patients undergoing stroke rehabilitation. Further, it took about 9600 kcal to gain 1 kg of body weight in these patients. Aggressive nutrition therapy is important for improving nutritional status and function in patients with malnutrition and sarcopenia.

## Figures and Tables

**Figure 1 nutrients-13-03274-f001:**
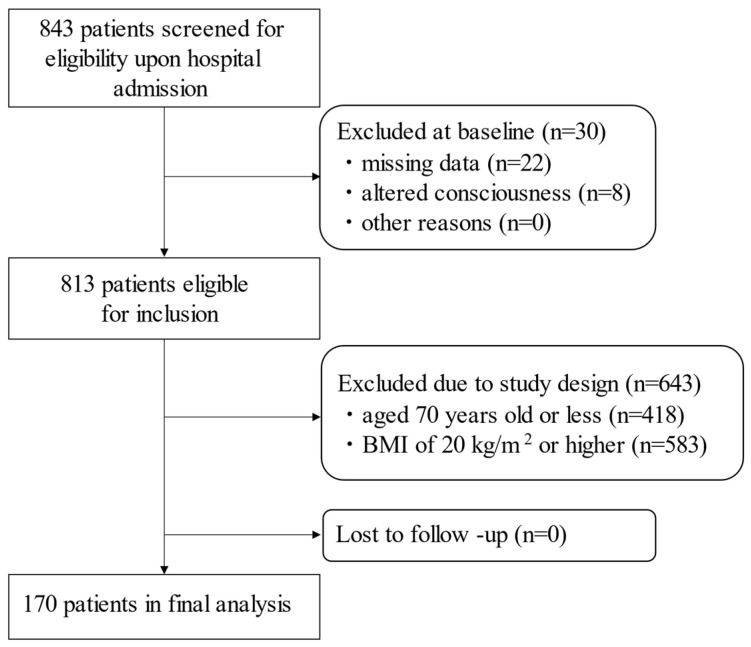
Flowchart of participant screening, inclusion criteria, and follow-up.

**Table 1 nutrients-13-03274-t001:** Baseline patient characteristics and between-group characteristics of patients with and without stored energy intake.

	Total (*n* = 170)	Patient without Stored Energy	Patient with Stored Energy	*p*
		(*n* = 56)	(*n* = 114)	
Age, *y*, mean (SD)	83.1 (6.2)	84.9 (6.4)	82.1 (5.9)	0.005
Sex (male), *n* (%)	52 (30.6)	22 (39.3)	30 (26.3)	0.111
Stroke type, *n* (%)				
Cerebral infarction	114 (67.1)	46 (82.1)	68 (59.6)	0.506
Cerebral hemorrhage	42 (24.7)	8 (14.3)	34 (29.8)	0.037
SAH	12 (7.1)	0 (0.0)	12 (10.5)	0.009
Stroke history, *n* (%)	46 (27.1)	18 (32.1)	28 (24.6)	0.359
Pre-stroke mRS, score, median (IQR)	1 (0,3)	0 (0,3]	1 (0,2)	0.876
Onset-admission days	12 (9,19)	12 (10,15)	14 (9,24)	0.291
Paralysis, *n* (%)				
Right/Left/Both	64 (37.6)/76 (44.7)/8 (4.7)	22 (39.3)/26 (46.4)/2 (3.6)	42 (36.8)/50 (43.9)/6 (5.3)	0.866
BRS, median (IQR)	5 (2,6)	3 (1,5)	5 (3,5)	
Upper limb	5 (2,6)	2 (1,5)	5 (3,5)	0.001
Hand-finger	5 (2,6)	3 (1,5)	5 (3,5)	<0.001
Lower limb				0.014
FIM, *score*, median (IQR)				
Total	49 (25,84)	39 (22,69)	59 (26,86)	0.034
Motor	32 (14,58)	20 (13,53)	37 (15,59)	0.033
Cognitive	17 (10,25)	16 (8,25)	18 (11,26)	0.232
Swallowing status				
FILS, score, median (IQR)	7 (6,9)	7 (2,7)	7 (7,9)	0.004
Dysphagia, *n* (%)	44 (25.9)	22 (39.3)	22 (19.3)	0.009
CCI, median (IQR)	3 (2,4)	3 (2,4)	3 (2,4)	0.687
MNA-SF, median (IQR)	4 (3,6)	3 (2,6)	4 (3,6)	0.047
Body composition, median (IQR]				
BW, kg	41.0 (38.2, 44.1)	43.6 (40.9, 47.5)	39.8 (38.0, 42.2)	<0.001
BMI, kg/m^2^	18.3 (16.7, 19.3)	18.4 (17.4, 19.4)	18.3 (16.5, 19.2)	0.268
SMM, kg	10.9 (8.8, 13.2)	12.6 (9.4, 15.1)	10.1 (8.8, 12.2)	<0.001
SMI, kg/m^2^	4.9 (4.0, 5.7)	5.2 (4.1, 6.1)	4.6 (4.0, 5.3)	0.002
FM, kg	9.8 (8.0, 13.5)	11.1 (8.3, 15.1)	9.5 (7.2, 12.9)	0.021
FMI, kg/m^2^	4.3 (3.3, 6.2)	4.4 (3.3, 6.5)	4.2 (3.2, 6.0)	0.295
HG, kg, median (IQR)	12.5 (5.6, 17.8)	14.4 (4.5, 20.6)	12.3 (6.0, 15.1)	0.199
Sarcopenia, *n* (%)	144 (84.7)	46 (82.1)	98 (86.0)	0.506
Total drug number, median (IQR)	5 (3, 8)	6 (4, 8)	5 (3, 7)	0.032
Laboratory data, mean (SD)				
Albumin, g/dL	3.28 (0.46)	3.3 (0.3)	3.3 (0.4)	0.503
Hemoglobin, g/dL	12.12 (1.40)	12.3 (1.4)	12.0 (1.3)	0.167
C-reactive protein, mg/dL	1.8 (2.7)	1.9 (2.4)	1.6 (2.9)	0.447

BMI, body mass index; BRS, Brunnstrom Recovery Stage; BW, body weight; CCI, Charlson’s Comorbidity Index; FILS, Food Intake Level Scale; FIM, Functional Independence Measure; FM, fat mass; FMI, fat mass index; HG, handgrip strength; MNA-SF, Mini Nutritional Assessment-Short Form; mRS, modified Rankin Scale; SAH, subarachnoid hemorrhage; SMI, skeletal muscle mass index; SMM, skeletal muscle mass.

**Table 2 nutrients-13-03274-t002:** Univariate analyses for outcomes between groups of patients with and without stored energy intake by stroke type.

	Cerebral Infarction			Cerebral Hemorrhage			Subarachnoid Hemorrhage		
	without Stored Energy	with Stored Energy	*p*	without Stored Energy	with Stored Energy	*p*	without Stored Energy	with Stored Energy	*p*
	(*n* = 46)	(*n* = 68)		(*n* = 8)	(*n* = 34)		(*n* = 0)	(*n* = 12)	
Energy intake									
On admission,	28.0 (26.0, 29.7)	35.3 (33.4, 37.9)	<0.001	28.4 (25.9, 29.0)	35.0 (31.5, 40.6)	<0.001	−	31.6 (31.1, 39.2)	−
kcal/kg/day									
On admission,	1200 (1200, 1200)	1400 (1400, 1400)	<0.001	1350 (1125, 1525)	1400 (1200, 1600)	0.426	−	1300 (1200, 1500)	−
kcal/day									
Set energy,	1299 (1208, 1410)	1205 (1101, 1263)	<0.001	1449 (1323, 1575)	1170 (1140, 1323)	0.005	−	1158 (1150, 1240)	−
kcal/kg/day									
Stored energy,	−126 (–214, –64)	195 (140, 299)		−76 (176, –50)	200 (60, 308)		−	67 (46, 262)	−
kcal/day									
On discharge,	31.4 (26.7, 38.3)	36.7 (35.4, 40.8)	<0.001	30.7(27.4, 34.4)	39.0 (34.7, 42.4)	0.001	−	32.8 (29.5, 38.4)	−
kcal/kg/day									
Protein intake									
g/kg/day									
On admission	1.1 (1.0, 1.2)	1.3 (1.2, 1.5)	<0.001	1.2 (1.1, 1.3)	1.3 (1.1, 1.5)	0.158	−	1.3 (1.1, 1.6)	−
On discharge	1.1 (1.0, 1.1)	1.3 (1.1, 1.4)	<0.001	1.2 (1.1, 1.2)	1.2 (1.1, 1.4)	0.001	−	1.2 (1.0, 1.6)	−
Change in BC									
kg									
BW	0.6 (0.0, 2.8)	0.3 (−0.9, 2.5)	0.101	−0.3 (−1.2, 0.7)	1.3 (−0.1, 3.0)	0.123	−	1.6 (0.5, 2.9)	−
SMM	0.3 (−0.1, 0.5)	0.3 (−0.4, 0.7)	0.947	−0.1 (−0.4, 0.0)	0.3 (0.1, 1.5)	0.002	−	0.7 (0.3, 1.8)	−
FM	1.1 (−1.2, 1.9)	0.5 (−1.0, 1.7)	0.721	−0.5 (−1.4, −0.0)	0.2 (−1.0, 0.8)	0.229	−	1.3 (−0.5, 2.8)	−
Changes in HG	0.1 (−1.5, 3.1)	1.3 (0.0, 5.1)	0.146	3.6 (−1.8, 4.3)	3.5 (0.0, 7.5)	0.503	−	2.2 (0.5, 6.3)	−
kg									
FIM gain									
total	21 (8, 34)	23 (16, 37)	0.337	38 (16, 57)	31 (5, 53)	0.247	−	27 (1, 29)	−
motor	19 (3, 29)	19 (12, 27)	0.495	30 (13, 48)	29 (3, 42)	0.248	−	19 (1, 23)	−
cognitive	3 (1, 6)	5 (2, 7)	0.182	5 (3, 7)	4 (0, 7)	0.244	−	6 (1, 9)	−
FILS on	8 (7, 10)	9 (7, 10)	0.418	9 (8, 9)	9 (7, 10)	0.84	−	8 (5, 9)	−
discharge									
Length of stay,	94 (68, 134)	83 (57, 124)	0.636	148 (136, 159)	91 (76, 118)	0.001	−	114 (58, 174)	−
day									
Rehabilitation	8 (7, 8)	8 (7, 8)	0.331	8 (8, 8)	8 (7, 8)	0.129	−	7 (5, 8)	−
units/day									

BC, body composition; BW, body weight; FILS, Food Intake Level Scale; FIM, Functional Independence Measure; FM, fat mass; HG, handgrip strength; SMM, skeletal muscle mass.

**Table 3 nutrients-13-03274-t003:** Multivariate regression analysis for BW gain during hospitalization.

	B (95% CI)	SE	β	*p*
Age, y	−0.036 (−0.109, 0.036)	0.037	−0.091	0.326
Sex (male)	0.466 (−0.508, 1.440)	0.493	0.087	0.346
Length of stay, day	0.011 (0.000, 0.023)	0.006	0.193	0.052
BRS-Lower limb	−0.160 (−0.406, 0.085)	0.124	−0.130	0.199
FIM-motor on admission	−0.021 (−0.055, 0.014)	0.017	−0.179	0.233
FIM-cognitive on admission	−0.029 (−0.094, 0.036)	0.033	−0.101	0.382
HG on admission, kg	0.132 (0.063, 0.201)	0.035	0.452	<0.001
Rehabilitation, units/day	−0.526 (−0.848, −0.204)	0.163	−0.275	0.002
Stored energy, kcal/day	0.003 (0.001, 0.005)	0.001	0.256	0.005
FMI on admission, kg/cm^2^	0.298 (0.069, 0.527)	0.116	0.227	0.011
SMI on admission, kg/cm^2^	−0.142 (−0.628, 0.344)	0.246	−0.064	0.564
Stroke type				
Cerebral infarction	0.035 (−1.387, 1.457)	0.72	0.007	0.961
Cerebral hemorrhage	−0.115 (−1.676, 1.445)	0.79	−0.020	0.884
SAH (control)	−	−	−	−

BRS, Brunnstrom Recovery Stage; BW, body weight; FIM, Functional Independence Measure; FMI, fat mass index; HG, handgrip strength; SAH, subarachnoid hemorrhage; SMI, skeletal muscle mass index.

**Table 4 nutrients-13-03274-t004:** Multivariate regression analysis for SMM gain during hospitalization.

	B (95% CI)	SE	β	*p* Value
Age, y	−0.045 (−0.105, 0.015)	0.03	−0.160	0.136
Sex (male)	0.576 (−0.056, 1.208)	0.318	0.166	0.074
Length of stay, day	−0.002 (−0.010, 0.007)	0.004	−0.047	0.676
BRS-Lower limb	0.074 (−0.109, 0.258)	0.092	0.09	0.424
FIM-motor on admission	−0.008 (−0.035, 0.019)	0.014	−0.111	0.55
FIM-cognitive on admission	−0.004 (−0.053, 0.046)	0.025	−0.020	0.881
HG on admission, kg	0.095 (0.036, 0.155)	0.03	0.478	0.002
Rehabilitation, units/day	−0.196 (−0.405, 0.014)	0.105	−0.168	0.066
Stored energy, kcal/day	0.002 (0.000, 0.004)	0.001	0.263	0.011
FMI on admission, kg/cm^2^	0.034 (−0.141, 0.209)	0.088	0.038	0.702
SMI on admission, kg/cm^2^	−1.275 (−1.628, −0.922)	0.178	−0.871	<0.001
Stroke type				
Cerebral infarction	−0.518 (−1.485, 0.448)	0.487	−0.150	0.29
Cerebral hemorrhage	−0.347 (−1.444, 0.749)	0.552	−0.090	0.531
SAH (control)	−	−	−	−

BRS, Brunnstrom Recovery Stage; BW, body weight; FIM, Functional Independence Measure; FMI, fat mass index; HG, handgrip strength; SAH, subarachnoid hemorrhage; SMI, skeletal muscle mass index.

## Data Availability

The data are not publicly available owing to opt out restrictions. Data sharing is not applicable.

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
