# Peer review of "Stored Energy Increases Body Weight and Skeletal Muscle Mass in Older, Underweight Patients after Stroke"

_nutrients, 2021, doi:10.3390/nu13093274_

Round 1

Reviewer 1 Report

For older and sarcopenia patients, appropriate nutrition support is a critical issue. this study emphasized the importance of nutrition support, and aim to evaluate how many calories needed to build body weight. 

Although there are no differences in BW gain and the SMM gain between the patient with stored energy intake and without energy intake, authors reported the stored energy intake associated with gain in BW and SMM. 

As authors mentioned, the nutrition therapy should be performed in combination with exercise, therefore, this article would become more helpful for physician if authors could describe about how they encourage these patients to take 1400kcal/kg/day. 

Author Response

Reviewer 1

For older and sarcopenia patients, appropriate nutrition support is a critical issue. this study emphasized the importance of nutrition support, and aim to evaluate how many calories needed to build body weight.

Although there are no differences in BW gain and the SMM gain between the patient with stored energy intake and without energy intake, authors reported the stored energy intake associated with gain in BW and SMM.

As authors mentioned, the nutrition therapy should be performed in combination with exercise, therefore, this article would become more helpful for physician if authors could describe about how they encourage these patients to take 1400kcal/kg/day.

(Response)

Thank you for the positive feedback. We agree that it is important that interventions including multidisciplinary nutrition therapy, exercise therapy, and pharmacotherapy provide patients with low nutrition with stored energy intake for weight gain. Accordingly, we revised the relevant sentences as follows.

(Change)

(4th paragraph, Discussion) “Ensuring the underweight older patients obtain the required level of stored energy is a practical nutritional management. This may also be true for underweight older patients in general, not just stroke patients. Aggressive rehabilitation nutrition improves nutrition-al and functional outcomes in patients with malnutrition and sarcopenia [8]. Malnutrition and sarcopenia negatively affect functional recovery and ADL, while nutrition improvement, including weight gain and muscle mass gain, is positively associated with functional recovery [11,12]. Aggressive nutrition therapy described here is characterized by the setting of energy requirements based on the amount energy expenditure per day plus stored energy. This is not the same as enhanced nutrition therapy and instead focuses on improving malnutrition and sarcopenia. Moreover, rehabilitation nutrition care process suggests that aggressive nutrition therapy should be performed in combination with aggressive exercise therapy [39]. Medication is another factor that can affect the nutritional status in older patients undergoing rehabilitation. Since some of the most important drugs that affect nutritional status are those that cause anorexia, rehabilitation pharmacotherapy should be practiced [33]. Therefore, it is necessary to reconcile the rehabilitation goal setting, the content, quantity, and quality of physical activity and exercise therapy, and the patient’s general condition to set nutrition goals through multidisciplinary collaboration.

Reviewer 2 Report

It is very interesting and important topic with practical value.

Remarks:

  1. You should mention something about pharmacological treatment especially statins that can impact sarcopenia.
  2. you should better describe convalescent rehabilitation program.It was long program. Is it 1 season or 3 times a day? Did you use any physical therapy, or speech therapy? Oral care and dysphagia rehabilitation should be also describe more precisely.
  3. Registration number of the study is missing.

Author Response

Reviewer 2

It is very interesting and important topic with practical value.

(Response)

Thanks for the positive feedback. Based on your comments, we have revised the manuscript as follows.

Remarks:

  1. You should mention something about pharmacological treatment especially statins that can impact sarcopenia.

(Response)

We agree that we should mention something about pharmacological treatment, because medication, including statins, is another factor that can affect the nutritional status of older patients undergoing rehabilitation. We accordingly revised the relevant sentences as follows.

(Change)

(2nd paragraph, Discussion) Stored energy intake was associated with gain in weight and muscle mass during hospitalization in geriatric rehabilitation after stroke. Considering the fact that weight gain and increased muscle mass have a positive impact on the ADLs of these patients [11,12], our findings suggest the need for enhanced nutritional management for older re-habilitation patients with low body weight. However, strong evidence exists that a large proportion of stroke patients do not even meet their estimated energy requirement both in hospital and after discharge [35]. An increase in energy demand related to rehabilitation, chronic diseases, and improvement of sarcopenia and malnutrition would likely result in negative energy balance in geriatric rehabilitation. Therefore, it is important to ensure the provision of individualized stored energy for weight gain in addition to energy requirements, while energy calculations should take into account post-stroke paralysis, muscle spasticity, physical disabilities, nutritional status, age and sex in geriatric rehabilitation after stroke. Furthermore, attention also needs to be paid to medications that are often pre-scribed to post-stroke patients. For example, statins are myotoxic and have been suggested to be associated with the progression of sarcopenia [36]. On the other hand, there is evidence that statin use reduces the incidence of sarcopenia, so if a patient is a good candidate for statin use, it is necessary to take care of myopathy and sarcopenia while using statins [37].”

(4th paragraph, Discussion) “Ensuring the underweight older patients obtain the required level of stored energy is a practical nutritional management. This may also be true for underweight older patients in general, not just stroke patients. Aggressive rehabilitation nutrition improves nutrition-al and functional outcomes in patients with malnutrition and sarcopenia [8]. Malnutrition and sarcopenia negatively affect functional recovery and ADL, while nutrition improvement, including weight gain and muscle mass gain, is positively associated with functional recovery [11,12]. Aggressive nutrition therapy described here is characterized by the setting of energy requirements based on the amount energy expenditure per day plus stored energy. This is not the same as enhanced nutrition therapy and instead focuses on improving malnutrition and sarcopenia. Moreover, rehabilitation nutrition care process suggests that aggressive nutrition therapy should be performed in combination with aggressive exercise therapy [39]. Medication is another factor that can affect the nutritional status in older patients undergoing rehabilitation. Since some of the most important drugs that affect nutritional status are those that cause anorexia, rehabilitation pharmacotherapy should be practiced [33]. Therefore, it is necessary to reconcile the rehabilitation goal setting, the content, quantity, and quality of physical activity and exercise therapy, and the patient’s general condition to set nutrition goals through multidisciplinary collaboration.

  1. you should better describe convalescent rehabilitation program. It was long program. Is it 1 season or 3 times a day? Did you use any physical therapy, or speech therapy? Oral care and dysphagia rehabilitation should be also describe more precisely.

(Response)

We agree. To describe convalescent rehabilitation in more detail and to add the description of oral management, swallowing rehabilitation as well as pharmacotherapy, we revised the relevant sentences as follows.

(Change)

(2.5. Convalescent rehabilitation, Materials and Methods) “The convalescent rehabilitation program (up to 3 hours per day) were performed according to the guidelines of the National Health Insurance System. The program was tailored to suit the functional abilities and disabilities of the patient, such as physical therapy including paralyzed limb facilitation (for leg paralysis), range of motion exercises, basic movement training (mainly for the legs), walking training, resistance training (such as chair-stand exercises [28]), and ADL training [29].

For nutritional management, nutritional screening and nutritional assessment were conducted for all patients, and under the guidance of dietitians and nutrition support team, active nutritional support was provided, including high-energy and/or high-protein meals. In addition, nutrition management was tailored to each patient's condition and nutritional needs by adjusting energy and protein according to changes in rehabilitation time and load [30].

Dysphagia rehabilitation was customized to the patients’ swallowing abilities and function, and included oral management and exercise, indirect (without food) and di-rect (with food) exercises, and diet modification through multi-occupational collaboration with speech and swallowing therapists, dental hygienists, and ward staff [31].

Oral management included oral screening, assessment, education, counseling, treatment (oral care), oral and dysphagia rehabilitation, medical treatment by a dentist, and practicing in cooperation with a multidisciplinary team [31]. Ward dental hygienists conducted oral and dysphagia rehabilitation, including indirect and direct (oral intake) exercises at patient bedside [32].

Medication management was carried out by multidisciplinary teams including pharmacists. Pharmacotherapy is one of the factors that affect the nutritional state of older people. Polypharmacy and inappropriate medications were corrected, and medications that could affect nutritional status were managed throughout the hospital stay [33].

  1. Registration number of the study is missing.

(Response)

Thanks for the comment. We have provided the IRB approval number as follows. On the other hand, due to its retrospective study design, we did not register the study protocol in advance.

(2.8. Ethics, Materials and Methods) “This study was approved by the Institutional Review Board (approval no.: 169-210617) of the hospital where the study was conducted. Written informed consent could not be obtained because of the constraints imposed by the retrospective study design, although the participants could withdraw from this study at any time by using an opt-out procedure. This study was conducted in accordance with the Declaration of Helsinki and ethical guidelines for medical and health research involving human sub-jects.

(Institutional Review Board Statement, just before References) “This study was approved by the Institutional Review Board (approval no.: 169-210617) of the hospital where the study was conducted. This study was con-ducted in accordance with the Declaration of Helsinki and ethical guidelines for medical and health research involving human sub-jects.

Reviewer 3 Report

This is an interesting study. This means about 9,600 kcal were needed to 28 gain 1 kg of BW. In addition, a 1-kg increase in body weight resulted in a 23.7% increase in SMM 29 and a 45.8% increase in FM.

However, the stroke patients over 70 years old and had a BMI of less than 20.0 kg/m2 received convalescent rehabilitation hospitalization for mean hospital stay of 100 ± 42 days, mean age was 83.1±6.2, more female, lower albumin with mean 3.28± 0.46 g/dL and higher C-reactive protein with 1.8± 2.7 mg/dL.

Patients with stored energy intake suffered from more cerebral hematoma and SAH. It needs to balance the study group and compared group.

The study selected with selection bias with older stroke patients, lower BMI and albumin. The group of stored energy intake suffered from more hemorrhagic stroke.

Multiple logistic regression analysis assessed the relationship and was better understood.

The daily stored energy intake was based on the estimation at the time of admission and was not based on accurate energy intake or changes in body weight over the entire hospitalization period. While the patients gain weight, the energy intake without change might influence the result.

Author Response

Reviewer 3

This is an interesting study. This means about 9,600 kcal were needed to 28 gain 1 kg of BW. In addition, a 1-kg increase in body weight resulted in a 23.7% increase in SMM 29 and a 45.8% increase in FM.

However, the stroke patients over 70 years old and had a BMI of less than 20.0 kg/m2 received convalescent rehabilitation hospitalization for mean hospital stay of 100 ± 42 days, mean age was 83.1±6.2, more female, lower albumin with mean 3.28± 0.46 g/dL and higher C-reactive protein with 1.8± 2.7 mg/dL.

(Response)

We appreciate your positive feedback. Based on your comments, we have revised the manuscript as follows.

  1. Patients with stored energy intake suffered from more cerebral hematoma and SAH. It needs to balance the study group and compared group.

(Response)

We agree. Because of the difference in stored energy between the two groups in patients with subarachnoid hemorrhage, the type of stroke should be controlled. Therefore, we modified the multiple linear regression analysis for two outcomes, BW change and SMM change, and adjusted for stroke type. According to the sample size, we also ensured that the number of confounding variables to be adjusted was appropriate. After reanalysis, our findings were the same. Accordingly, we revised the relevant sentences as follows.

(Change)

(2.7. Statistical analysis, Materials and Methods) “The values were reported as the mean (SD) for parametric data or as the median [interquartile range; IQR] and numbers (%) for non-parametric and categorical data, respectively. In the univariate analyses, the patients were stratified according to the intake of stored energy (with or without stored energy). Comparisons between groups were made using the t-test, Mann-Whitney U test, and chi-square test, as suitable.

Multiple linear regression analysis was carried out to determine whether the stored energy was independently associated with the study outcomes, including BW gain and SMM gain during hospitalization. The covariates of age, sex, length of hospital stay, BRS of the lower limb, FIM-motor and FIM-cognitive scores at admission, total rehabilitation therapy (units/day), energy intake at baseline, FMI and SMI on admission, stroke type, all of which are considered to be confounding factors affecting BW gain, were included in the model. Multicollinearity was assessed using the variance inflation factor (VIF): a VIF value of 1–10 indicated the absence of multicollinearity. Statistical significance was set at P-values of <0.05. All analyses were conducted using IBM SPSS version 21 (Armonk, NY, USA).

(4th paragraph, Results) “In the multivariate linear regression analysis, the stored energy intake was significantly and positively associated with BW gain during hospitalization (b = 0.256, p = 0.005), after adjusting for potential confounders, including age, sex, baseline energy intake, FMI and SMI (Table 3). Moreover, the stored energy intake was significantly and positively as-sociated with SMM gain during hospitalization (b = 0.263, p = 0.011) in the same analysis model (Table 4).

Table 3 and 4 have been updated.

  1. The study selected with selection bias with older stroke patients, lower BMI and albumin.

(Response)

Thanks for the comment. In this study, our interest was to examine the stored energy for weight gain and muscle mass gain in underweight older adults. For this purpose, we defined the target population as those over 70 years old and with a BMI of less than 70, based on the GLIM criteria, which is the global consensus for diagnosing underweight (Cederholm, T., et al. Clin Nutr. 2019). As a result, the subjects were older, had a lower BMI, and had lower albumin levels. Of course, it is difficult to generalize the results of this study as it is, so future validation with subjects of various backgrounds is needed. Accordingly, this study limitations have been revised in the discussion as follows.

(Change)

(5th paragraph, Discussion) “…Further, this study involved a single rehabilitation hospital in Japan, possibly limiting the generalizability of the findings. In addition, there was a selection bias in that the subjects were older hospitalized patients with low body weight. Future multicenter studies are needed to determine whether similar results can be obtained in diverse populations. Lastly, due to its retrospective study design, we were unable to obtain detailed information on nutritional therapy during hospitalization, possibly affecting the results. High-quality prospective studies adjusted for confounders are needed in the future.

  1. Multiple logistic regression analysis assessed the relationship and was better understood.

(Response)

Thanks for the comment. We agree that it might be easier to understand if multiple logistic regression analysis was used to evaluate the relationship between stored energy and the outcomes. On the other hand, it may not be appropriate to use the present outcomes (weight gain and muscle mass gain) as binary variables with validated cutoff values. Therefore, in this study, we treated the outcomes as quantitative variables and used multiple linear regression analysis for the study results.

  1. The daily stored energy intake was based on the estimation at the time of admission and was not based on accurate energy intake or changes in body weight over the entire hospitalization period. While the patients gain weight, the energy intake without change might influence the result.

(Response)

Thanks for the comment. We completely agree with you. This is one of the major study limitations that we must acknowledge. We also understand that it would be ideal to closely monitor weight changes during hospitalization, measure energy needs over time using indirect calorimetry, and carefully adjust the amount of energy provided. Therefore, future high-quality research is needed to address this point. However, because of the lack of evidence on the relationship between stored energy and weight gain and muscle mass gain in geriatric rehabilitation patients, we believe that it is worthwhile to present this study with an understanding of the limitation and that this study will contribute to the development of clinical nutrition. We have clearly stated this study limitation in the manuscript as follows.

(5th paragraph, Discussion) “This study had some limitations. First, the daily stored energy intake was based on estimates at the time of admission and not on weight change over the course of the hospital stay. This could be a major limitation of this study, as energy adjustments are made as appropriate for weight changes over time in daily clinical practice. Moreover, determination of energy requirements by indirect calorimetry over time would provide more accurate estimates than those used in this study. Further, this study involved a single rehabilitation hospital in Japan, possibly limiting the generalizability of the findings….

Round 2

Reviewer 2 Report

Current version of the paper titled Stored energy increases body weight and skeletal muscle mass in older, underweight patients after stroke is much improved. In my opinion it is accepted in current form.

Author Response

Authors’ Response

Dear editors and reviewers in Nutrients. We sincerely appreciate your positive comments in the previous review, which really helped us improve our manuscript. Based on the comments received, we have done our best to make additional corrections in this revised manuscript. Our changes have marked in red in the revised manuscript.

Reviewer 2

Current version of the paper titled Stored energy increases body weight and skeletal muscle mass in older, underweight patients after stroke is much improved. In my opinion it is accepted in current form.

(Response)

Thanks for the positive comment.

Reviewer 3

The study groups showed super older patients and lower BMI.

The patients without stored energy intake are older than patients with stored energy intake.

The linear regression showed correlation rather than association. The stroke type of stored energy intake group must be compared with non-energy store group.

(Response)

Thanks for the supportive comment, which will further improve the quality of our manuscript. We agree that a two-group analysis with and without stored energy by stroke type should be performed. We updated our results by comparing the two groups of outcomes by stroke type in our reanalysis. As a result, in subarachnoid hemorrhage, all patients were assigned to the group with stored energy, presumably due to the small sample size. Because we adjusted for stroke type in the multivariate analysis, our conclusions remain the same, but we thought we should state them clearly in the study limitations. Accordingly, we revised the relevant manuscript, including Results, Discussion, and Table 2, as follows.

(Change)

(3rd paragraph, Results) “Overall, the study patients gained an average of 1.0 ± 2.6 kg of BW over a mean hospital stay of 100.2 ± 42.3 days, with a mean stored energy intake of 96.2 ± 91.4 kcal per day. They also gained an average of 0.2 ± 1.6 kg of SMM and 0.5 ± 2.3 kg of FM, respectively. This means that it took about 9,600 kcal (9,614.2) for the patient to gain 1 kg of BW (mean stored energy per day [96.2] × mean number of days in hospital [100.2]/mean BW increase [1.0]). In addition, a 1-kg increase in body weight resulted in a 23.7% increase in SMM (mean SMM increase ÷ mean BW increase × 100) and a 45.8% increase in FM (mean FM increase ÷ mean BW increase × 100). Table 2 shows a two-group comparison of the nutritional profiles of patients with and without stored energy intake by stroke type. Except for patients with subarachnoid hemorrhage (SAH), patients with stored energy intake of more than 0 kcal/day consumed more energy and protein on admission and at discharge than those with stored energy intake of less than 0 kcal/day. All patients with SAH were supplemented with stored energy. In the univariate analysis, there was no difference in BW gain or SMM gain between the two groups.

(2nd paragraph, Discussion) “Stored energy intake was associated with gain in weight and muscle mass during hospitalization in geriatric rehabilitation after stroke. The results of the univariate analysis showed differences by stroke type, but the small sample size of 12 patients with SAH might have affected the results. On the other hand, since we adjusted for stroke type in the multivariate analysis, we consider that the effect of stroke type on the results was controlled. Considering the fact that weight gain and increased muscle mass have a positive impact on the ADLs of these patients [11,12], our findings suggest the need for enhanced nutritional management for older rehabilitation patients with low body weight. However, strong evidence exists that a large proportion of stroke patients do not even meet their estimated energy requirement both in hospital and after discharge [35] …...

(Last paragraph, Study limitation section in Discussion) “This study had some limitations. First, the daily stored energy intake was based on estimates at the time of admission and not on weight change over the course of the hospital stay. This could be a major limitation of this study, as energy adjustments are made as appropriate for weight changes over time in daily clinical practice. Moreover, determination of energy requirements by indirect calorimetry over time would provide more accurate estimates than those used in this study. Further, this study involved a single rehabilitation hospital in Japan, possibly limiting the generalizability of the findings. In addition, there was a selection bias in that the subjects were older hospitalized patients with low body weight. Future multicenter studies are needed to determine whether similar results can be obtained in diverse populations, especially by stroke type….”

Table 2 have been updated to show the result by stroke type.

Reviewer 3 Report

The study groups showed super older patients and lower BMI.

The patients without stored energy intake are older than patients with stored energy intake.

The linear regression showed correlation rather than association. The stroke type of stored energy intake group must be compared with non-energy store group.

Author Response

(The authors gave the same response as above.)
